# AgenticNet: Rethinking Multi-agent System Architectures with LLM-based Networks

## Abstract

Multi-agent systems (MAS) enhance the capabilities of single LLM agents by leveraging collaboration and specialization. However, existing designs often rely on ad-hoc coordination strategies, lacking a principled architecture that integrates reasoning, communication, and adaptation. This limitation makes it difficult to scale multi-agent systems in a way that is both effective and interpretable. To address this challenge, we rethink MAS design through a structured layered topology, which we call AgenticNet. In this framework, lower layers act as planners that decompose problems, intermediate layers function as executors that advance reasoning step by step, and upper layers serve as synthesizers that verify consistency and deliver final decisions. Information propagates forward through the layers, while adaptation is guided by layer-level prompt updaters and a global prompt supervisor that refine agent behavior based on task loss, serving as a heuristic feedback propagation mechanism conceptually inspired by backpropagation. We conduct extensive experiments on five benchmarks, including AIME24, GSM8K, MATH500, HumanEval, and MBPP. AgenticNet achieves the highest average score across five benchmarks (85.56%), with strong gains on code generation (HumanEval 98.8%, MBPP 98.7%), while remaining competitive with the best methods on math reasoning.

## 1 Introduction

Large language models (LLMs) have demonstrated remarkable capabilities in reasoning (Wei et al., 2022), problem solving (Wei et al., 2022), and tool use (Schick et al., 2023), but single-agent deployments often struggle with scalability, robustness, and specialization. To address these limitations, recent research has explored multi-agent systems (MAS) (Park et al., 2023), where multiple LLM agents collaborate through interaction and role differentiation. Such systems can enhance performance by dividing labor, integrating diverse skills, and enabling emergent behaviors beyond those achievable by a single agent.

Existing MAS architectures can be broadly categorized into four types. Equi-level structures (Du et al., 2023) place all agents at the same hierarchy, encouraging collective decision-making but lacking mechanisms to resolve conflicts or enforce coherent strategies. Hierarchical structures (Hong et al., 2024a) rely on leader–follower dynamics, which simplify coordination but introduce single points of failure and reduce system robustness. Nested or hybrid structures (Li et al., 2023b) combine equi-level and hierarchical sub-systems, providing flexibility but greatly increasing design complexity and making agent interactions harder to control. Dynamic structures (Liu et al., 2024) allow agents to adapt roles based on context, which improves flexibility but makes behaviors unstable and difficult to interpret. The typical MAS architectures are shown in Fig. 1.

Despite their differences, these paradigms share several fundamental limitations. First, planning remains largely ad hoc, as current systems lack principled mechanisms to decompose complex tasks and coordinate contributions across heterogeneous agents. Second, memory and adaptation are weak: MAS struggle to maintain long-term consistency across extended reasoning horizons and lack systematic methods to refine agent behavior when errors accumulate. Third, scalability and interpretability remain major obstacles, since interactions among agents become increasingly difficult to control as the number of agents grows, and collective behavior often appears opaque and hard to audit. Consequently, while these categories demonstrate

the diversity of MAS design, existing approaches remain fragmented and lack a principled architectural foundation.

To overcome these limitations, we propose AgenticNet, a network topology conceptually inspired by layered architectures that provides a principled design for multi-agent systems. Building on the analogy of LLM-as-computational-unit explored in prior work such as LMNet (Wu et al., 2025), we organize agents into layered structures and introduce a natural-language-based forward–feedback training paradigm. Lower layers act as planners that decompose tasks, intermediate layers function as executors that advance reasoning step by step, and upper layers serve as synthesizers that verify consistency and produce final outputs. To address memory and adaptation, we introduce layer-level prompt updaters and a global prompt supervisor that iteratively refine agent behavior based on task loss through a heuristic feedback propagation mechanism conceptually inspired by backpropagation. While LMNet uses dense-vector communication and differentiable optimization, AgenticNet operates entirely in natural language, making it applicable to black-box LLM APIs. Our textual feedback propagation further extends TextGrad's text-gradient concept (Yuksekgonul et al., 2024) to a structured multi-layer agent optimization setting with a two-level prompt update hierarchy. To improve scalability and interpretability, the layered design of AgenticNet provides modularity and transparency, allowing reasoning chains to be structured, auditable, and easier to analyze. The AgenticNet architecture is shown in Fig. 2.

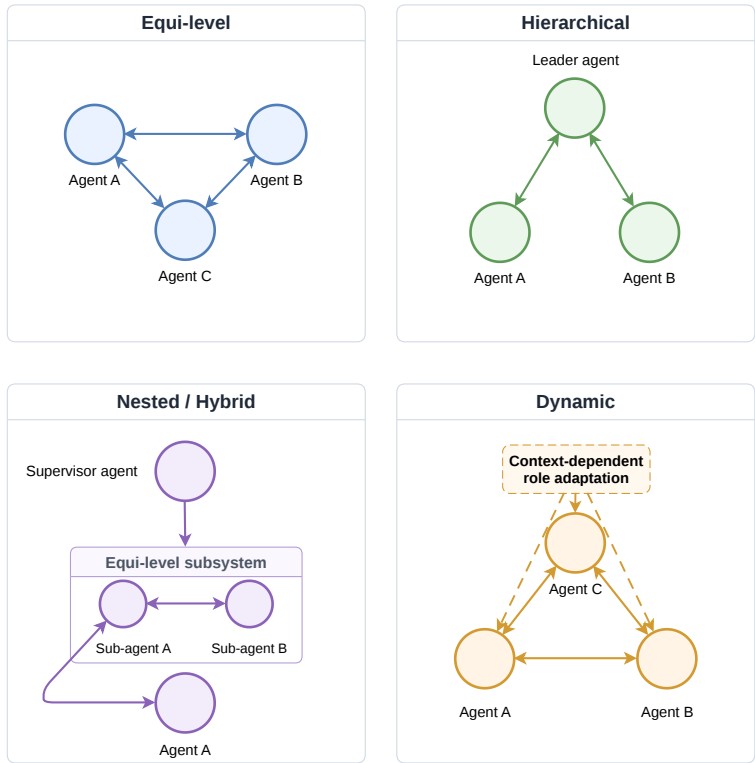

Figure 1: Representative architectures of typical multi-agent systems.

## 2 Related Work

### 2.1 Multi-Agent System Architectures.

LLM-based agent research has evolved from single-agent designs to increasingly sophisticated multi-agent systems (MAS). Early approaches such as Chain-of-Thought and ReAct Wei et al. (2022); Yao et al. (2023)

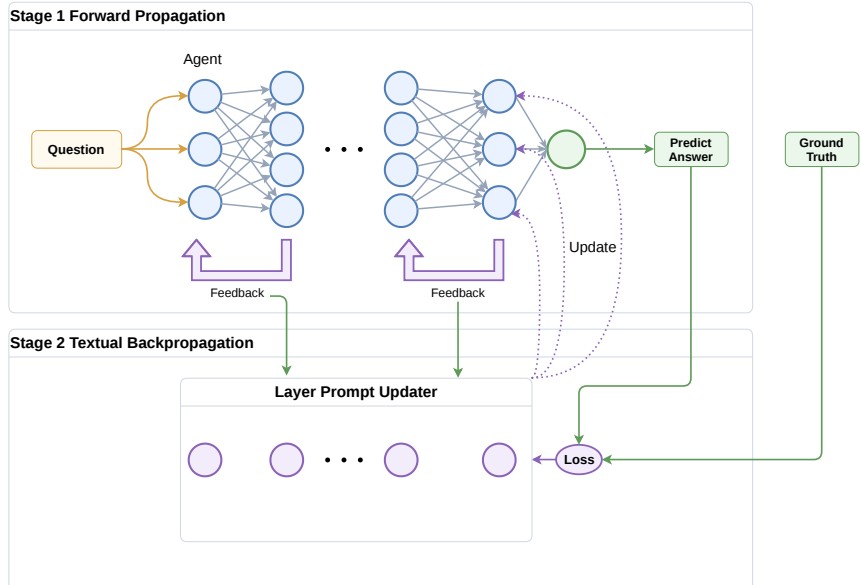

Figure 2: The architecture and optimization mechanism of AgenticNet. The framework operates through a dual-stage process. In Forward Propagation (Stage 1), the input query traverses a layered network where agents function as computational nodes. Agents process upstream information by applying preference ranking and Top-K context selection before generating intermediate reasoning steps. When the final prediction diverges from the ground truth during the optimization stage, Textual Feedback Propagation (Stage 2) is initiated. A task-specific loss activates Layer Prompt Updaters, which analyze the discrepancy together with the forward-pass trace to synthesize natural-language feedback for correcting the reasoning logic. A Rewrite operator then applies these feedback signals to iteratively refine the agents' system prompts.

rely on a single LLM for reasoning and acting, while MAS improve capability by distributing roles across multiple interacting agents Wang et al. (2025). Existing MAS are typically either general-purpose architectures with static or dynamic coordination Wu et al. (2024); Li et al. (2023a); Hong et al. (2024b); Chen et al. (2023b) or application-specific systems tailored to domains such as code generation, scientific discovery, and e-commerce Qian et al. (2023); Huang et al. (2023); Ghafarollahi & Buehler (2025; 2024); Li et al. (2023c); Zhao et al. (2023).

## 2.2 Self-evolving multi-agent systems

Self-evolving MAS dynamically adjust prompts, workflows, or agent topologies during execution Hu et al. (2024); Zhang et al. (2025b); Yuan et al. (2024). Representative examples include DyLAN Liu et al. (2024), EvoFlow Zhang et al. (2025a), GPTSwarm, AgentSquare, and AFlow Zhuge et al. (2024); Shang et al. (2024); Zhang et al. (2024). In contrast, AgenticNet adopts a fixed layered architecture with loss-driven prompt updates, aiming for a design that is easier to analyze, audit, and scale.

## 2.3 LLM Networks and Textual Optimization

LMNet Wu et al. (2025) proposes organizing LLMs as vertices in a neural-network-like graph with dense-vector communication and differentiable optimization. AgenticNet shares the architectural inspiration of structuring multiple LLM components into a graph, but differs in both communication and optimization: it uses natural-language messages rather than dense vectors, making it compatible with black-box API models, and it updates prompts through natural-language feedback rather than differentiable parameter updates. TextGrad Yuksekgonul et al. (2024) introduces textual gradients, namely natural-language feedback that plays a role analogous to numerical gradients during optimization. AgenticNet builds on this idea by

80 applying text-based feedback within a layered multi-agent forward graph and by introducing a two-level
81 prompt optimization hierarchy with Layer Prompt Updaters for per-agent revisions and a Global Prompt
82 Updater for cross-layer coordination.

## 3 Methodology

84 We present AgenticNet, a multilayer multi-agent architecture that adopts a feed-forward topology conceptu-
85 ally inspired by multi-layer structures while operating entirely in text space. AgenticNet formalizes agents
86 as discrete nodes whose parameters are natural-language prompts; layers of such agents compose into a
87 feed-forward computation graph, and adaptation is performed through natural-language feedback. Below
88 we describe the architecture and the learning procedure.

### 3.1 Architecture

90 Let $q$ denote an input query and $y^\star$ the ground-truth answer. AgenticNet is composed of $L$ layers indexed
91 by $l = 1, \ldots, L$; layer $l$ contains $N^{(l)}$ agents. The prompt of the $i$-th agent in layer $l$ is $p_i^{(l)}$. The model before
92 training is $f_\theta(\cdot)$. We denote the set of textual outputs produced by layer $l$ as

$$X^{(l)} = \{x_i^{(l)}\}_{i=1}^{N^{(l)}}, \tag{1}$$

93 where each $x_i^{(l)}$ is the output of agent $i$ in layer $l$.

94 The topmost layer contains a dedicated Synthesizer agent with prompt $p^{(\text{final})}$ that consumes $X^{(L)}$ and emits
95 a single consolidated prediction $\hat{y}$. A judge returns a dataset-dependent vector of diagnostic signals

$$\mathbf{L} = (L_1, \ldots, L_m), \qquad L_t \in \mathcal{C}_t, \tag{2}$$

96 where $m$ and the semantics of each component $L_t$ (e.g., reasoning validity, arithmetic correctness, answer
97 correctness, format compliance) are chosen according to the target task. In the simplest case $\mathcal{C}_t = \{0, 1\}$.
98 These signals are combined into a scalar training objective by a weighted sum,

$$\mathcal{L}(\hat{y}, y^\star) = \sum_{t=1}^{m} w_t \big(1 - \ell_t(\hat{y}, y^\star)\big), \tag{3}$$

99 where $\ell_t(\hat{y}, y^\star)$ denotes the judge-provided score (equal to $L_t$ for binary criteria) and $w_t \geq 0$ are validation-
100 selected weights.

### 3.2 Forward propagation

102 We denote the input wrapper as $X^{(0)} = \{\text{Question: } q\}$. For each layer $l = 1, \ldots, L$ and each agent
103 $i \in \{1, \ldots, N^{(l)}\}$, the forward computation proceeds by letting the agent rank all upstream messages
104 $\{x_j^{(l-1)}\}_{j=1}^{N^{(l-1)}}$ according to its own preference. A compact context is then constructed by selecting the
105 top-$k$ messages in the ranked list,

$$\mathcal{J}_i^{(l)} = \text{TopK}\left(\{x_j^{(l-1)}\}_{j=1}^{N^{(l-1)}}, k\right), \qquad \tilde{x}_i^{(l)} = x_{j_1}^{(l-1)} \oplus x_{j_2}^{(l-1)} \cdots \oplus x_{j_k}^{(l-1)}, \tag{4}$$

106 where indices $j_1, \ldots, j_k$ follow the descending order in the agent's ranking and $\oplus$ denotes textual concate-
107 nation under a fixed prompt template that specifies the agent's role. The agent then generates its output
108 conditioned on this context as

$$x_i^{(l)} = f_\theta\big(p_i^{(l)} \oplus \tilde{x}_i^{(l)} \oplus q\big). \tag{5}$$

109 The Synthesizer collates and analyzes $X^{(L)}$ under $p^{(\text{Synthesizer})}$, then directly generates a consolidated pre-
110 diction $\hat{y}$.

### 3.3 Feedback Propagation and Prompt Refinement

The feedback propagation stage in AgenticNet is not numerical gradient backpropagation as used in standard neural networks. Instead, adaptation is driven by natural-language feedback estimates produced by large language models. The core idea is to use the scalar loss $\mathcal{L}$ provided by the judge, together with the layer-wise intermediate outputs, to generate actionable revision suggestions for each agent prompt.

Concretely, at layer $l$, a Layer Prompt Updater inspects the current prompt set

$$P^{(l)} = \{p_i^{(l)}\}_{i=1}^{N^{(l)}}, \tag{6}$$

the upstream and downstream outputs $(X^{(l-1)}, X^{(l)})$, the ground-truth answer $y^\star$, and the overall loss $\mathcal{L}$. For each input-output pair $(x_j^{(l-1)}, x_j^{(l)})$, the Layer Prompt Updater generates a feedback message

$$f_j^{(l)} = F\big(x_j^{(l-1)}, x_j^{(l)}, y^\star, \mathcal{L}\big), \tag{7}$$

which provides actionable suggestions for improving the corresponding agent prompt. All feedback messages are collected into a set

$$\mathcal{F}^{(l)} = \{f_j^{(l)}\}_{j=1}^{|X^{(l)}|}. \tag{8}$$

Based on this set of feedback, the Layer Prompt Updater produces for each agent $i$ a textual modification suggestion

$$\Delta p_i^{(l)} = \text{Aggregate}\big(\mathcal{F}^{(l)}\big), \tag{9}$$

which indicates how $p_i^{(l)}$ should be changed to reduce $\mathcal{L}$. For analogy with iterative optimization, we write the conceptual update as

$$p_i^{(l)} \leftarrow p_i^{(l)} - \widehat{\nabla}_{p_i^{(l)}} \mathcal{L}, \tag{10}$$

where the subtraction symbol denotes applying the updater's suggested modification, and $\widehat{\nabla}_{p_i^{(l)}} \mathcal{L}$ is an abstract language-formatted update rather than a true numerical vector.

In practice, the prompt update is carried out by a Rewrite step: the current prompt and the suggested modifications are fed to a rewrite procedure that produces a candidate new prompt

$$p_i^{(l)\,\text{new}} = \text{Rewrite}\big(p_i^{(l)}, \Delta p_i^{(l)}\big). \tag{11}$$

Besides, a Global Prompt Updater is introduced to update the prompts of the Layer Prompt Updaters and ensure that layer-specific objectives are aligned with the overall task. Concretely, the Global Prompt Updater receives the prompt of the Layer Prompt Updater for layer $l$, $P^{(l)}$, together with the corresponding inputs and outputs of that layer, $X^{(l-1)}$ and $X^{(l)}$. Based on this information and the total loss $\mathcal{L}$, it generates a textual modification suggestion $\Delta P_{\text{global}}^{(l)}$. The final updated prompt for the layer is then obtained as

$$P^{(l)\,\text{new}} = \text{Rewrite}\Big(P^{(l)}, \Delta P_{\text{global}}^{(l)}\Big). \tag{12}$$

Through this mechanism, the Global Prompt Updater provides cross-agent and cross-output coordination within the layer.

## 4 Experiment

### 4.1 Experiment Setup

**Benchmarks.** We evaluate `AgenticNet` on five challenging public benchmarks that span mathematical reasoning and code generation. These include **GSM8K** Cobbe et al. (2021), a dataset of grade-school math word problems; **MATH** Hendrycks et al. (2021), which contains more difficult high-school competition-level math problems; **AIME2024** HuggingFaceH4 (2024), a set of challenging math competition problems;

HumanEval Chen et al. (2021) for code generation; and **MBPP** Austin et al. (2021) for Python code generation. For the MATH dataset, we selected a subset of 500 level-5 problems for the main comparison against other MAS.

**Baselines.** To provide a comprehensive evaluation, we compare our framework against 14 representative methods, which we categorize into three groups:

- **Single-Agent Methods:** standard prompting techniques such as CoT Wei et al. (2022), Complex-CoT Fu et al. (2023), and Self-Consistency (SC) Wang et al. (2023).

- **Hand-Crafted Multi-Agent Systems:** manually designed role and workflow systems, including MultiPersona Wang et al. (2024), LLM-Debate Du et al. (2023), LLM-Blender Jiang et al. (2023), AgentVerse Chen et al. (2023a), and MacNet Qian et al. (2025).

- **Automated Multi-Agent Systems:** frameworks that search for or adapt workflows, including DyLAN Liu et al. (2024), AutoAgents Chen et al. (2024), GPTSwarm Zhuge et al. (2024), ADAS Hu et al. (2024), AgentSquare Shang et al. (2024), and AFlow Zhang et al. (2024).

**Implementation Details.** Unless otherwise noted, all experiments use `gpt-4o-mini` with temperature 0.7 and a maximum output length of 4096 tokens. AgenticNet uses the $[2, 3, 3]$ architecture (three layers, eight agents) and two epochs of textual feedback propagation. Test sizes are $n = 1319$ for GSM8K, $n = 485$ for MATH500, $n = 75$–$90$ for AIME2024 depending on the filtered split, $n = 164$ for HumanEval, and $n = 427$ for MBPP. API pricing is \$0.15 per 1M input tokens and \$0.60 per 1M output tokens.

**Train/Test Split and Inference Setup.** For all ablation and cross-benchmark experiments, textual feedback propagation is performed on a held-out subset of 10–15 samples per benchmark. All reported accuracies are measured on disjoint test data under strict exact-match evaluation. The optimization stage and the benchmark evaluation stage are strictly separated: ground-truth answers are used only on the held-out training split to generate feedback and revise prompts. After optimization, the updated prompts are frozen. During benchmark evaluation, AgenticNet runs forward propagation only, and neither the Judge nor the ground-truth answers are available to the agents.

## 4.2 Performance Analysis

As shown in Table 1, AgenticNet achieves the highest average score (85.56%), driven by its performance on code generation (HumanEval 98.8%, MBPP 98.7%). On math reasoning, it is competitive with the strongest baselines: it is within 0.5 percentage points of MaAS on GSM8K (91.8% vs. 92.30%), while trailing diversity-based methods on MATH (49.03% vs. MermaidFlow's 55.42%). These results suggest that the layered architecture is particularly effective when problems benefit from explicit decomposition and synthesis, while harder mathematical reasoning remains more sensitive to model capacity and search diversity.

## 4.3 Ablation Study

To isolate the impact of textual feedback propagation, we compare the same AgenticNet $[2, 3, 3]$ architecture with and without training while keeping the backbone fixed to `gpt-4o-mini`. We also compare the trained AgenticNet directly against a vanilla baseline under the same strict exact-match protocol.

Table 2a shows that textual feedback propagation is most beneficial on MBPP (+21.2 points), while gains on MATH500 are modest and AIME2024 remains unchanged with only 15 held-out samples. HumanEval exhibits a small ceiling-effect regression from an already saturated zero-shot baseline. Table 2b further shows that AgenticNet improves over vanilla on all five benchmarks, with the largest gains on code generation.

## 4.4 Computational Cost Analysis

Table 3 reports average per-sample efficiency on the MATH level-5 subset under an identical evaluation protocol, including the same LossCalculator setting and four evaluation calls per sample. AgenticNet incurs

Table 1: **Performance comparison** among single-agent, hand-crafted, and automated multi-agent systems. All baselines use `gpt-4o-mini` as the base model. Best and second-best results are **bolded** and underlined, respectively.

| Method | GSM8K | MATH | HumanEval | MBPP | Avg. |
|---|---|---|---|---|---|
| Vanilla | 87.45 | 46.29 | 87.08 | 71.83 | 73.16 |
| CoT Wei et al. (2022) | 87.10 | 46.40 | 88.13 | 71.83 | 73.87 |
| ComplexCoT Fu et al. (2023) | 86.89 | 46.53 | 87.49 | 72.36 | 73.32 |
| SC (CoT×5) Wang et al. (2023) | 87.57 | 47.91 | 88.60 | 73.60 | 74.42 |
| MultiPersona Wang et al. (2024) | 87.50 | 45.43 | 88.32 | 73.19 | 73.61 |
| LLM-Debate Du et al. (2023) | 89.47 | 48.54 | 88.68 | 70.29 | 74.25 |
| LLM-Blender Jiang et al. (2023) | 88.35 | 46.92 | 88.80 | 77.05 | 75.28 |
| DyLAN Liu et al. (2024) | 89.98 | 48.63 | 90.42 | 77.30 | 76.58 |
| AgentVerse Chen et al. (2023a) | 89.91 | 47.35 | 89.29 | 74.28 | 75.21 |
| MacNet Qian et al. (2025) | 87.95 | 45.18 | 84.57 | 65.28 | 70.75 |
| AutoAgents Chen et al. (2024) | 87.69 | 45.32 | 87.64 | 71.95 | 73.15 |
| GPTSwarm Zhuge et al. (2024) | 89.14 | 47.88 | 89.32 | 77.43 | 75.94 |
| ADAS Hu et al. (2024) | 86.12 | 43.18 | 84.19 | 68.13 | 70.41 |
| AgentSquare Shang et al. (2024) | 87.62 | 48.51 | 89.08 | 78.46 | 75.92 |
| AFlow Zhang et al. (2024) | 91.16 | 51.28 | 90.93 | 81.67 | 78.76 |
| MaAS Zhang et al. (2025c) | **92.30** | 51.82 | 92.85 | 82.17 | 82.25 |
| MermaidFlow Zheng et al. (2025) | 91.4 | **55.42** | 92.87 | 82.31 | 82.77 |
| **AgenticNet (Ours)** | 91.8 | 49.03 | **98.8** | **98.7** | **85.56** |

(a) Ablation study on the textual feedback propagation mechanism.

| Benchmark | w/o | w/ | Delta |
|---|---|---|---|
| HumanEval | 100.00% | 98.8% | -1.2 |
| MBPP | 77.52% | 98.7% | +21.2 |
| MATH500 | 70.15% | 71.6% | +1.5 |
| AIME2024 | 8.89% | 8.89% | 0.0 |

(b) Vanilla vs. AgenticNet under consistent evaluation.

| Benchmark | Vanilla | AgenticNet | Delta |
|---|---|---|---|
| GSM8K | 87.4 | 91.8 | +4.4 |
| MATH500 | 64.0 | 71.6 | +7.6 |
| AIME2024 | 6.67 | 8.89 | +2.2 |
| HumanEval | 87.0 | 98.8 | +11.8 |
| MBPP | 85.4 | 98.7 | +13.3 |

a clear efficiency cost relative to simpler baselines. Across all eight methods and 50 evaluation samples, the total experimental cost is $7.69; even the most expensive method, LLM-Debate, costs $0.0700 per sample, or roughly $70 per 1,000 problems. In this comparison, LLM-Blender offers the strongest accuracy-cost trade-off. AgenticNet uses about half as many calls as LLM-Debate (62 vs. 122.2 per sample) but remains more expensive than Vanilla CoT and LLM-Blender. More broadly, the results indicate that architecture matters at least as much as raw compute budget: DyLAN costs more than Vanilla CoT while performing worse, whereas LLM-Blender outperforms LLM-Debate despite being much cheaper. These results suggest that AgenticNet should be used with explicit latency-cost trade-offs in mind, even though computational cost is not itself a practical barrier at current API pricing.

## 4.5 Impact of Network Depth

To investigate the scalability of AgenticNet and identify the effect of architectural depth, we evaluate MATH performance with $L \in \{2, 3, 4, 5\}$ using two backbones: `gpt-4o-mini` and `DeepSeek-V3`. This comparison shows how model capacity interacts with the amount of layered coordination.

As shown in Figure 3 and Table 4, depth affects the two backbones differently. For `DeepSeek-V3`, performance increases overall with depth, rising from 77.6% at $L = 2$ to 82.4% at $L = 5$. This trend suggests that a

Table 3: Average per-sample efficiency metrics on the MATH level-5 subset (50 test samples, random state 42) using `gpt-4o-mini`.

| Method | Accuracy | Calls/Sample | Cost/Sample | Time/Sample (s) |
|---|---|---|---|---|
| VanillaCoT | 22.0% | 14.8 | $0.0036 | 4.9 |
| ADAS(3 iterations) | 18.0% | 12.0 | $0.0046 | 27.2 |
| DyLAN(6-agent pool, top-3) | 20.0% | 29.6 | $0.0098 | 17.3 |
| AgenticNet $[2, 3, 3]$(zero-shot) | 12.0% | 62.7 | $0.0152 | 17.4 |
| AgenticNet $[2, 3, 3]$(trained) | 24.0% | 62.3(+16.0 amortized) | $0.0311 | 22.8 |
| MultiPersona(8 personas) | 28.0% | 37.5 | $0.0185 | 27.4 |
| LLM-Debate(4 agents, 2 rounds) | 28.0% | 122.2 | $0.0700 | 90.1 |
| LLM-Blender(8 candidates, top-3 fusion) | 34.0% | 20.0 | $0.0074 | 35.0 |

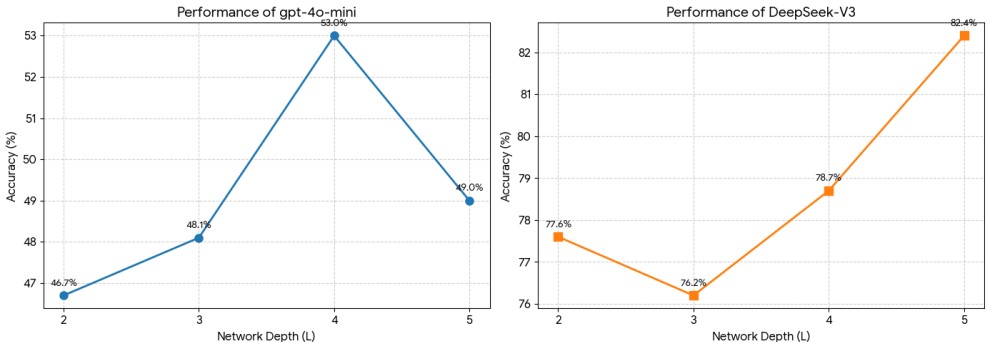

Figure 3: Performance of AgenticNet with `gpt-4o-mini` and `Deepseek-V3` backbone across different network depths.

stronger base model can better exploit deeper coordination patterns. By contrast, `gpt-4o-mini` follows a non-monotonic trajectory: performance improves from 46.7% at $L = 2$ to 53.0% at $L = 4$, then drops to 49.0% at $L = 5$, indicating that additional layers can eventually introduce more noise than benefit for a weaker backbone.

While the depth experiment shows that the optimal $L$ varies by base model, the architecture still uses a globally fixed depth for all queries within a given run. In practice, shallower configurations may suffice for simpler queries, and a fixed architecture can therefore incur unnecessary computational overhead on those cases. Dynamic depth routing and early-exit mechanisms are natural directions for future work.

### 4.6 Training Dynamics and Convergence Analysis

To examine the stability of textual feedback propagation, we track MATH accuracy across sequential training epochs using the `gpt-4o-mini` backbone. Here, one epoch denotes one feedback-and-rewrite cycle.

Figure 4 and Table 5 show rapid initial adaptation followed by stable convergence. Accuracy rises from 49.0% at Epoch 1 to 52.7% at Epoch 3, then remains stable at 52.1% and 52.5% in Epochs 4 and 5. This pattern suggests that the main gains are realized early and that the learned prompts remain robust afterward.

To investigate whether textual feedback propagation learns reusable reasoning patterns or instead overfits to the training distribution, we train AgenticNet on 15 held-out samples from MATH-500 or AIME 2024 for one epoch and evaluate on the other dataset. As shown in Table 6, training on AIME 2024 improves MATH-500 from 71.6% to 75.80%, while training on MATH-500 degrades AIME 2024 from 8.89% to 5.56%. This asymmetry suggests that harder-problem training transfers better than easier-problem training.

Table 4: Performance comparison on the MATH dataset with varying network depths ($L$). Accuracy is reported in percentage (%).

| Base Model | $L = 2$ | $L = 3$ | $L = 4$ | $L = 5$ |
|------------|---------|---------|---------|---------|
| gpt-4o-mini | 46.7 | 48.1 | **53.0** | 49.0 |
| DeepSeek-V3 | 77.6 | 76.2 | 78.7 | **82.4** |

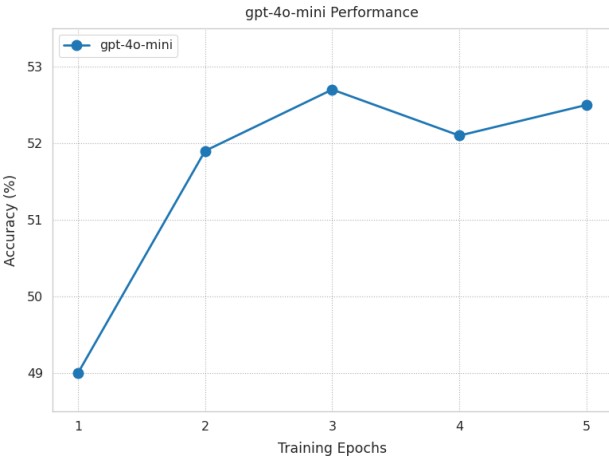

Figure 4: Learning trajectory of AgenticNet with the `gpt-4o-mini` backbone over five training epochs.

## 4.7 Impact of Network Width

To evaluate the horizontal scalability of AgenticNet, we investigate the impact of network width $W$ on reasoning performance. Here, width $W$ is defined as the number of parallel agents in the Executor and Synthesizer layers. We conduct experiments on the MATH dataset using the `gpt-4o-mini` backbone, varying $W \in \{1, 2, 3, 4\}$ while keeping the first layer fixed.

As shown in Table 7, widening the architecture yields modest but non-monotonic gains. The model reaches 49.0% at both $W = 1$ and $W = 3$, dips slightly to 48.3% at $W = 2$, and peaks at 50.9% at $W = 4$. These results suggest that additional parallel agents can improve robustness by exposing the Synthesizer to a broader set of reasoning traces, but the gains are not purely linear because wider coordination also introduces more noisy intermediate states.

## 4.8 Heterogeneous Model Assignment

While the main experiments use a homogeneous model backbone where all agents share the same LLM, `AgenticNet` natively supports per-role model selection since each agent independently instantiates its LLM. Different roles—Planners, Executors, and Synthesizers—may naturally benefit from different model capabilities: Planners may benefit from strong reasoning models, Executors from models with reliable arithmetic, and Synthesizers from models with robust consistency-checking ability. To investigate this hypothesis, we conduct a heterogeneous model assignment study on the MATH-500 benchmark in the zero-shot setting. We use the $[2, 3, 3]$ architecture and fix `gpt-4o-mini` as the base backbone, then selectively replace individual roles or all roles with either `GPT-4.1-mini`, a stronger API model, or `Qwen3.5-Flash`, a locally deployed model with strong reasoning capability.

As shown in Table 8, the results reveal a pronounced role asymmetry. Upgrading only the Synthesizer yields the largest single-role accuracy gain (+14 to +16 percentage points over baseline), nearly matching the performance of an all-role upgrade. This suggests that the final answer extraction and error-correction step is the most quality-sensitive role in the pipeline, and that Planner and Executor outputs already contain sufficient signal even when produced by the weaker `gpt-4o-mini` backbone. By contrast, upgrading only

Table 5: Evolution of MATH accuracy across training epochs using the `gpt-4o-mini` backbone.

| Metric | E1 | E2 | E3 | E4 | E5 |
|---|---|---|---|---|---|
| Accuracy (%) | 49.0 | 51.9 | **52.7** | 52.1 | 52.5 |

Table 6: Cross-dataset transfer in the math domain. Training uses 15 held-out samples for one epoch with the `gpt-4o-mini` backbone and the $[2, 3, 3]$ architecture.

| Direction | Single Model | AgenticNet (w/o) | AgenticNet (w/) |
|---|---|---|---|
| MATH-500 → AIME 2024 | 5.56% | 8.89% | 5.56% |
| AIME 2024 → MATH-500 | 64.0% | 71.6% | 75.80% |

the Executor provides no benefit and even degrades relative to baseline ($-2$ pp), indicating that a stronger Executor cannot compensate for downstream synthesis errors when the Synthesizer remains weak.

The results also highlight the cost-efficiency of targeted role-specific model selection. Replacing only the Synthesizer with `Qwen3.5-Flash` achieves 88.00% accuracy, while replacing all three roles with `GPT-4.1-mini` achieves the same 88.00% at a comparable cost but with different latency profiles. In contrast, the homogeneous all-`Qwen3.5-Flash` configuration reaches 92.00%, the highest overall accuracy, demonstrating that a capable locally deployed model can outperform both heterogeneous and homogeneous API-based configurations. These findings confirm that heterogeneous, role-aware model assignment is both supported and beneficial in `AgenticNet`, and that cost-aware routing—assigning stronger models to quality-sensitive roles (Synthesizer) while retaining cheaper models for input-processing roles (Planner, Executor)—is a promising direction for practical deployment.

### 4.9 Case Study

To demonstrate the end-to-end operational flow and adaptive learning capabilities of AgenticNet, we present a qualitative case study from the MATH benchmark. This example illustrates how the network processes a complex query, how an arithmetic error propagates through the layered pipeline, and how feedback-based prompt refinement identifies and corrects the underlying logic.

#### 4.9.1 Problem Definition

Consider the input query $q$ from the MATH benchmark: "The graph of $y = ax^2 + bx + c$ is a parabola with a vertical axis of symmetry. The vertex of this parabola is $(2, 3)$ and the parabola contains the point $(4, 4)$. Find the value of $y$ when $x = 6$." The ground-truth answer is $y^* = 7$.

#### 4.9.2 Forward Propagation and Error Manifestation

In the forward pass (Figure 5), the query is processed by a three-layer network. In Layer 1, the system generates a coherent decomposition, denoted as $X^{(1)}$, which is then passed to Layer 2. The executor agents correctly derive the vertex form $y = a(x - 2)^2 + 3$ and solve for the coefficient $a = \frac{1}{4}$.

Table 7: Performance scaling of AgenticNet relative to network width $W$. Results are reported for the MATH reasoning benchmark.

| Width ($W$) | Accuracy (%) |
|---|---|
| $W = 1$ | 49.0 |
| $W = 2$ | 48.3 |
| $W = 3$ | 49.0 |
| $W = 4$ | **50.9** |

Table 8: **Heterogeneous model assignment on MATH-500** (50 test samples, zero-shot). $\Delta$ is measured against the homogeneous `gpt-4o-mini` baseline (72.00%). Best and second-best results are **bolded** and underlined, respectively.

| Configuration | Planner (L0) | Executor (L1) | Synthesizer (L2) | Accuracy | $\Delta$ |
|---|---|---|---|---|---|
| All `Qwen3.5-Flash` | `Qwen3.5-Flash` | `Qwen3.5-Flash` | `Qwen3.5-Flash` | **92.00%** | **+20.00** |
| `Qwen3.5-Flash`@Synth | `gpt-4o-mini` | `gpt-4o-mini` | `Qwen3.5-Flash` | 88.00% | +16.00 |
| All `GPT-4.1-mini` | `GPT-4.1-mini` | `GPT-4.1-mini` | `GPT-4.1-mini` | 88.00% | +16.00 |
| `GPT-4.1-mini`@Synth | `gpt-4o-mini` | `gpt-4o-mini` | `GPT-4.1-mini` | 86.00% | +14.00 |
| `GPT-4.1-mini`@Planner | `GPT-4.1-mini` | `gpt-4o-mini` | `gpt-4o-mini` | 76.00% | +4.00 |
| Baseline (all `gpt-4o-mini`) | `gpt-4o-mini` | `gpt-4o-mini` | `gpt-4o-mini` | 72.00% | – |
| `GPT-4.1-mini`@Executor | `gpt-4o-mini` | `GPT-4.1-mini` | `gpt-4o-mini` | 70.00% | -2.00 |

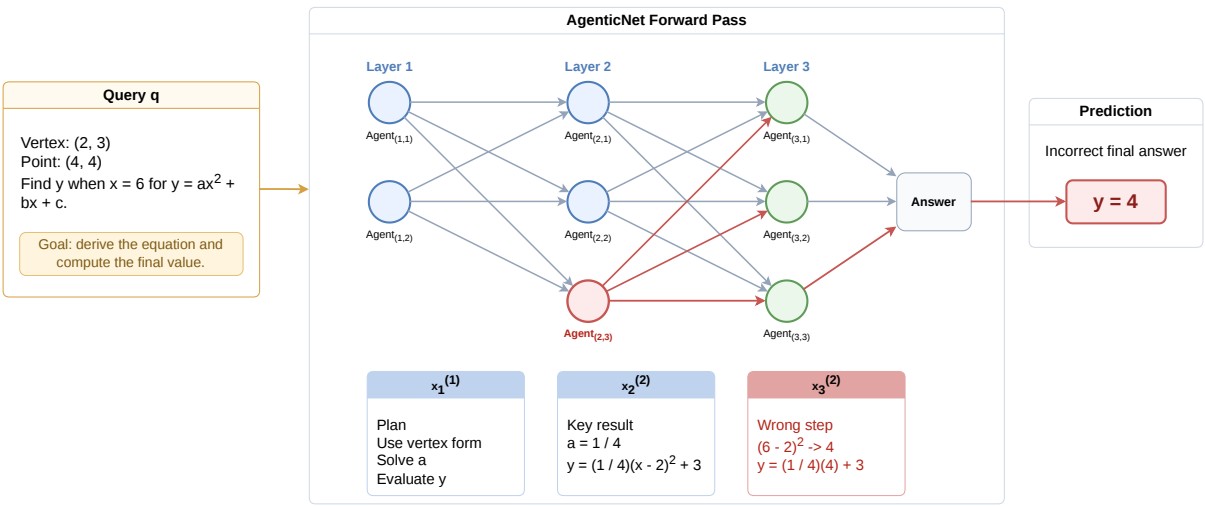

Figure 5: The Forward Propagation stage for the case study. The query $q$ is processed by layer 1 and layer 2. A calculation error (highlighted in red) originates in agent $x_3^{(2)}$, which propagates through layer 3 to produce an incorrect final answer $y = 4$.

However, in Layer 2, agent $x_3^{(2)}$, which is responsible for the final numerical evaluation, introduces a critical arithmetic oversight by evaluating $4^2$ as 4 rather than 16. This flawed intermediate result propagates to Layer 3, where the error goes undetected, leading to the incorrect final prediction $y = 4$.

### 4.9.3 Textual Feedback Propagation and Optimization

The feedback propagation stage (Figure 6) is triggered when the Judge identifies a discrepancy between the predicted output $y = 4$ and the ground-truth value $y^* = 7$. This non-zero task loss $\mathcal{L}$ activates the Layer Prompt Updater for Layer 2.

The LPU inspects the forward-pass data $(X^{(1)}, X^{(2)})$ alongside the ground truth, identifies the error source in agent $x_3^{(2)}$, and generates a targeted feedback message $f_3^{(2)}$: "The execution of Step 5 is incorrect. The expression $(6 - 2)^2$ simplifies to $4^2$, which equals 16. Your calculation incorrectly evaluated this as 4. The correct calculation is $y = \frac{1}{4}(16) + 3 = 7$."

This feedback is processed by the `Rewrite` procedure, which produces a modification $\Delta p_3^{(2)}$ to update the agent prompt $p_3^{(2)}$. The updated prompt includes a preventative rule: *When evaluating mathematical expressions, verify exponentiation operations before simplification.*

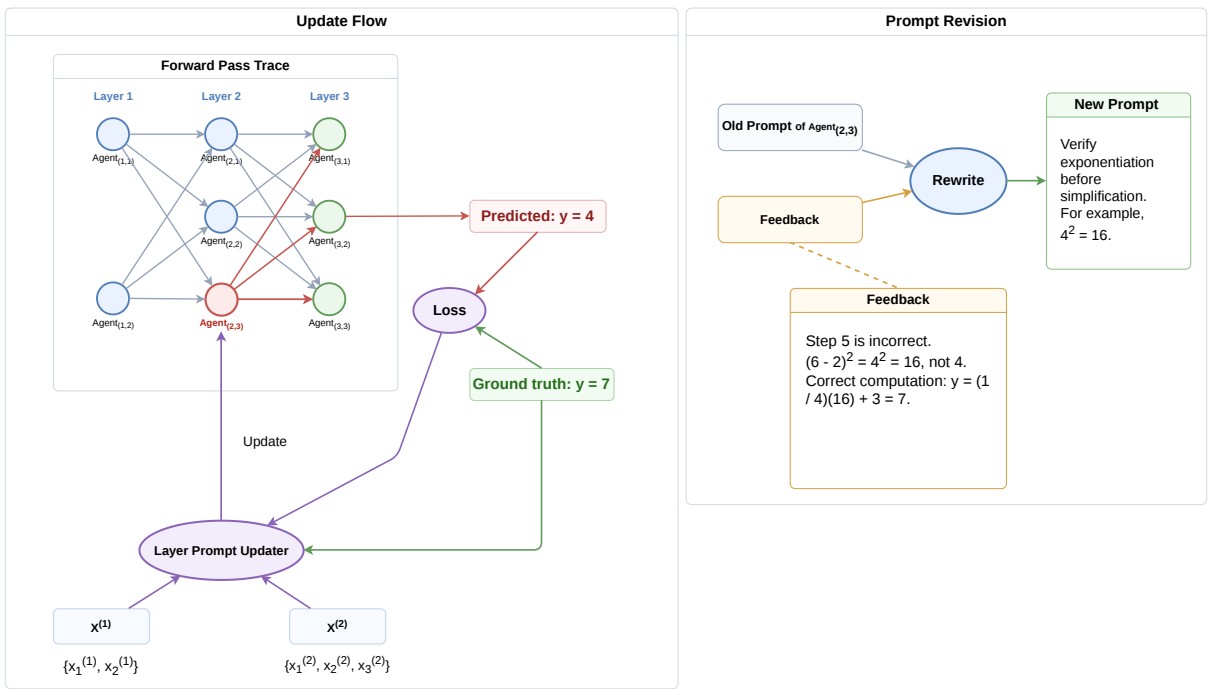

Figure 6: The Textual Feedback Propagation stage for the case study. The non-zero loss $\mathcal{L}$ activates the Layer Prompt Updater (LPU). The LPU inspects the forward-pass trace, identifies the error source $(x_3^{(2)})$, and generates a specific feedback message $(f_3^{(2)})$. This feedback is used by the `Rewrite` procedure to update the agent prompt $p_3^{(2)}$ to $p_3^{(2)\text{new}}$, correcting the logic for future tasks.

## 5 Conclusion

In this paper, we introduced AgenticNet, a layered multi-agent framework for more structured multi-agent design. AgenticNet uses fixed roles for planning, execution, and synthesis, together with textual feedback propagation for prompt refinement. Experiments show the highest average performance (85.56%), driven by strong code-generation results and competitive math reasoning performance.

Additional analyses clarify the strengths and limitations of the framework. Cross-dataset transfer suggests that training on harder problems can yield reusable reasoning heuristics, although the learned prompts remain partially distribution-dependent. Heterogeneous model assignment reveals a strong role asymmetry: upgrading the Synthesizer yields the largest gains, while Executor-only upgrading degrades performance, motivating cost-aware per-role routing. The current architecture also uses a globally fixed depth for all queries within a run, making adaptive per-query depth an important future direction.

**Broader Impact.** AgenticNet's textual feedback propagation produces human-auditable prompt updates, offering greater transparency than opaque numerical fine-tuning. However, the framework could still be misused for harmful objectives, and automated prompt refinement may overfit to narrow metrics at the expense of broader safety considerations. We therefore recommend prompt auditing and safety evaluation before deployment in sensitive domains.

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
